# Enteral Nutrition in Pediatric Crohn’s Disease: New Perspectives

**DOI:** 10.3390/nu17193124

**Published:** 2025-09-30

**Authors:** Viviana Fara Brindicci, Rosangela Grieco, Roberta Giusy Ruiz, Sabrina Cardile, Teresa Capriati, Chiara Maria Trovato, Giulia Bolasco, Daniela Knafelz, Fiammetta Bracci, Arianna Alterio, Francesca Ferretti, Domenica Elia, Elena Spinetti, Ruggiero Francavilla, Antonella Diamanti

**Affiliations:** 1Digestive Diseases and Nutritional Rehabilitation Unit, Nutritional Treatments of Complex Diseases Research Unit, Bambino Gesù Children’s Hospital, IRCCS, 00165 Rome, Italy; rosangela.grieco@uniroma1.it (R.G.); robertagiusy.ruiz@opbg.net (R.G.R.); sabrina.cardile@opbg.net (S.C.); teresa.capriati@opbg.net (T.C.); chiaramaria.trovato@opbg.net (C.M.T.); giulia.bolasco@opbg.net (G.B.); daniela.knafelz@opbg.net (D.K.); fiammetta.bracci@opbg.net (F.B.); arianna.alterio@opbg.net (A.A.); francesca.ferretti@opbg.net (F.F.); domenica.elia@opbg.net (D.E.); antonella.diamanti@opbg.net (A.D.); 2Interdisciplinary Department of Medicine, Pediatric Section, Children’s Hospital ‘Giovanni XXIII’, University of Bari “Aldo Moro”, 70126 Bari, Italy; ruggiero.francavilla@uniba.it; 3Pediatric Gastroenterology and Liver Unit, Maternal and Child Health Department, Sapienza-University of Rome, 00185 Rome, Italy; 4Department of Paediatrics, Università Cattolica del Sacro Cuore, IRCCS, 00168 Rome, Italy

**Keywords:** enteral nutrition, Crohn’s disease, pediatric, dysbiosis, fiber, inflammation

## Abstract

**Background/Objectives**: The efficacy of exclusive enteral nutrition (EEN) on the induction of remission of Crohn’s disease (CD) has been demonstrated with different diets (elemental, semi-elemental, and polymeric). A narrative review was conducted to assess the effects of different enteral diets in pediatric CD patients, considering the hypothesis that manipulating the nutritional key ingredients may enhance the clinical efficacy. **Methods**: An extensive literature search was performed across PubMed, Embase, and the Cochrane Library, covering all records published up to 27 July 2025. Both pediatric and adult studies were considered, and nutritional composition was compared with remission rates. **Results**: Twelve studies involving patients with active CD treated with EEN were found. Most studies were conducted with polymeric diets (*n* = 8), which achieved a high remission rate (up to 85%), thus confirming their advantage over other EEN diets. **Conclusions**: EEN with polymeric diets satisfies the need to revert the acute inflammation in most pediatric CD patients. Polymeric formulas have two advantages: (a) they contain transforming growth factor-β (TGF-β), which exerts anti-inflammatory effects on intestinal epithelial cells, and (b) they have a mixed-fat composition, including saturated fatty acids (SFAs), monounsaturated fatty acids (MUFAs), polyunsaturated fatty acids (PUFAs) as well medium-chain triglycerides (MCTs), which provides better results than EEN diets enriched with single-fat components. However, pathophysiological evidence shows gut microbiota alterations after EEN begins, despite clinical improvement. So, a potential strategy to enhance the efficacy of polymeric diets may be fiber enrichment.

## 1. Introduction

Exclusive enteral nutrition (EEN) consists of a complete liquid diet that meets all nutritional requirements while excluding all regular solid foods for a duration of 6 to 8 weeks [1]. This type of nutritional intervention is recommended as induction therapy to achieve remission in Crohn’s disease (CD) [1]. In 1973, Voitk and coworkers [2] reported the first successful experience with EEN, administered as an exclusive elemental diet for an average of 22 days, in 12 of the 13 treated patients. The diet was well-tolerated and led to weight gain and reduced inflammation; moreover, two of the nine patients awaiting surgery no longer required it after the nutritional intervention. Currently, EEN in pediatric populations is considered the first-line treatment for mild-to-moderate CD according to the guidelines of the European Society for Paediatric Gastroenterology, Hepatology and Nutrition (ESPGHAN), demonstrating superiority over steroids in promoting mucosal healing [1]. Overall, EEN seems to induce remission in up to 85% of children with mild-to-moderate CD [3,4].

### 1.1. How Does EEN Work in CD?

CD patients frequently develop malnutrition due to reduced food intake, intestinal malabsorption, increased energy requirements, and chronic loss of protein [5]. Therefore, they may require EEN, which provides additional nutrients addressing malnutrition [6]. By contrast, parenteral nutrition (PN) is not considered the primary nutritional option in pediatric CD, although it is an appropriate way to support patients with a dysfunctional gastrointestinal tract [5].

In fact, there is evidence that EEN directly impacts intestinal inflammation [6]. Systemic inflammatory markers, such as C-reactive protein (CRP) and erythrocyte sedimentation rate (ESR), are normalized with EEN before any change in the nutritional status [7]. Thus, in CD, EEN can revert intestinal inflammation and the related malnutrition due to the blockage of catabolic effects induced by a cytokine storm [6].

Since the 1980s, it has been known that EEN has a direct anti-inflammatory effect on the intestine in CD, mediated by improved intestinal permeability. Sanderson et al. [8] studied intestinal permeability to sugar to assess the efficacy of an elemental diet as an exclusive treatment of the small bowel in CD. Fourteen children aged 11–17 years with active CD received an elemental diet for six weeks: tests with iso-osmolar oral test solutions before and after this treatment demonstrated abnormally elevated lactulose/rhamnose permeability ratios in all 14 children, which fell significantly after the EEN treatment [8]. This change overlapped with marked clinical improvement, as assessed by a disease activity index score [8].

Currently, the concept of intestinal permeability, recognized as a key trigger of inflammation in CD, refers to the disruption of homeostasis among mucosal, immunological, and microbiological barriers. This loss of homeostasis initiates inflammation in CD, and EEN acts at multiple levels to restore barrier function [5].

#### 1.1.1. Mucosal Barrier

The epithelial barrier is the basal layer of the mucosa: it comprises enterocytes and other specialized cell types. Between the epithelial cells, apical junctional complexes, which include tight junctions and adherens junctions, cooperate to protect the intestine against harmful macromolecules and microorganisms [9]. Under homeostatic conditions, the epithelial barrier provides an efficient physical, chemical, and electrical barrier. When the integrity of the epithelium is compromised, luminal antigens, including pathobionts, can access the lamina propria, where they are sampled by dendritic cells, thereby activating and/or sustaining deregulated inflammatory immune responses [5]. The crosstalk between dendritic cells and the mesenteric lymph nodes causes the activation of the inflammatory storm, which produces several cytokines (see below) responsible for the loss of integrity of epithelial cells and of tight junctions [5]. The epithelial layer is also covered by a dense mucus gel composed of an outer layer of secreted mucins lying above a compact inner glycocalyx, synthesized by the goblet cells and inaccessible to most bacteria [9]. The mucus layer plays a crucial role in intestinal homeostasis because reduced mucin secretion by goblet cells predisposes to inflammatory bowel disease (IBD) [10]. In addition to acting as a bio-physical barrier, mucus creates a matrix that permits the retention of high concentrations of antimicrobial molecules, such as lysozymes, defensins, cathelicidins, lipocalins, and C-type lectins, such as RegIIIγ5 [10]. These peptides are secreted in an inducible fashion by Paneth cells and interact with the microbiota, inhibiting colonization by pathogenic microbes in favor of commensal ones. There is substantial evidence that Paneth cell dysfunction and impaired defensin secretion may contribute to IBD susceptibility, as well as the colonization by specific bacteria, in particular adherent-invasive *Escherichia coli* (AIEC) [11]. Exclusive enteral nutrition seems to inhibit interleukin (IL)-1β, IL-6, IL-8, interferon γ (IFN-γ), and tumor necrosis factor α (TNF-α), thus reverting the effects of the cytokines on epithelial cells and tight junctions [12,13]. Enteral nutrition also reduces the richness of Enterobacteriaceae in the intestinal microbiological community, thereby reducing the trigger for inflammation associated with this class of pathobionts (see below). The enteral nutrition intervention can also have trophic effects on the epithelial barrier, as shown by the overrepresentation of genes involved in aromatic amino acid metabolism and spermidine/putrescine transport, which play an important role in cellular growth and may suggest increased epithelial cell renewal and tissue repair [14,15]. Therefore, EEN may increase the expression of tight junction proteins between epithelial cells, reversing the increased gut permeability seen in CD [16]. Bacteria and, in particular, AIEC strains are also hypothesized to play a pivotal role in regulating the permeability of the mucosal barrier (see below) [11]. *Akkermansia muciniphila* strains may also be involved in regulating mucosal barrier permeability: this is a mucin-degrading bacterium residing in the mucus layer, and it is crucial to control host mucus turnover, which, in turn, improves gut barrier functions [17,18]. So, EEN in CD patients can regulate the mucosal barrier by modulating the composition of the microbial ecosystem [19].

#### 1.1.2. Immunological Barrier

The intestinal lamina propria contains gut-associated lymphoid tissue (GALT), whose components include T-invariant cells, innate lymphoid cells, and plasmacytoid dendritic cells [20]. They cooperate as an extensive network with multifollicular lymphoid tissues like Peyer’s patches and isolated lymphoid follicles [20]. Under homeostatic conditions, both dendritic cells and macrophage populations have specific adaptations that promote tolerance [20,21]. During infection, immune responses shift toward a more inflammatory profile, which may result in immune-mediated pathology when dysregulated [21,22]. Then, GALT immune cells play a crucial role in maintaining a balance between tolerance to beneficial microbes and nutrients and protection against harmful pathogens, representing the immunological barrier [20,22]. A dysregulated innate immune response to luminal microbial or nutritional antigens, as well as an increased intestinal permeability, leads to the entry of antigens and pathogens into the lamina propria, where they are recognized by dendritic cells that present them to immune cells in the lymphatic compartment of Peyer’s patches and isolated lymphoid follicles [20,21]. The contact with dendritic cells and class 2 major histocompatibility complex (MHC) induces the differentiation of naïve CD4 T cells into T-helper (Th)-1, principally by the stimulation of IL-12 and IL-18 [20,21]. This contact triggers a proinflammatory immune response via the activation of Th-17, Th-1, and Th-2 and the inhibition of CD4^+^CD25^+^Foxp3^+^ T-regulatory lymphocytes (Tregs) [23,24]. Th1 and Th17 cells are involved in the activation of the cytokine cascade through the secretion of TNF-α, IFN-γ, IL-12, and IL-23 [25]. Furthermore, whereas CD4 T cells in the normal lamina propria undergo apoptosis, this process does not occur in T cells from CD patients, which are markedly resistant to apoptosis, thereby perpetuating inflammation [26]. Several cytokines in the inflammatory cascade are known to prevent T cell apoptosis, particularly IL-2, IL-15, IL-6, IL-12, and IL-18 [26]. The switch of CD4 T cells to Th1 is not balanced by Tregs [25,27], which are reduced in the blood during the acute phases of inflammation of CD; changes in the ratio between peripheral and mucosal Treg cells are indeed a sensitive marker for intestinal inflammation [28,29].

The lipid content of EEN, in particular, short-chain fatty acids (SCFAs), can also alter the gene expression of proteins, acting as a signal between the epithelial cell and other intestinal cells, including surface molecules, such as MHC class II, thus weakening the presentation of harmful antigens and the activation of adaptive immunity [6,30]. Exclusive enteral nutrition, as previously mentioned, suppresses the release of IFN-γ, TNF-α, IL-1β, IL-6, and IL-8. Furthermore, EEN significantly increases the relative and absolute numbers of Treg cells, likely by enhancing the activity of TGF-β, a cytokine that suppresses inflammation and promotes the development of Tregs [26,31].

#### 1.1.3. Microbiological Barrier

The microbiological barrier involves a complex gut microbial community, predominantly composed of five bacterial phyla: Firmicutes (60–80%), Bacteroidetes (20–40%), Verrucomicrobia, Actinobacteria, and Proteobacteria [5]. The general features of CD-associated dysbiosis include a reduction in Firmicutes and Verrucomicrobia and an expansion of the phylum Proteobacteria, as well as an overall decrease in bacterial diversity [32,33]. This imbalance is described as a pathogenetic contributor to IBD [14]. The depletion of Firmicutes, and, at the species level, the reduced abundance of *Faecalibacterium prausnitzii,* is a well-characterized feature of CD, likely reflecting the ability of this species to produce anti-inflammatory metabolites, including SCFAs [14]. Moreover, CD-associated dysbiosis has also been linked to the expansion of potentially pathogenic pathobionts [14]. In CD patients, indeed, a high prevalence of *E. coli* (Proteobacteria) is demonstrated by the high representation of KEGG modules coding for ubiquinone [15]. Ubiquinone is produced exclusively by organisms capable of aerobic respiration, such as *E. coli*, and its primary role is to maintain membrane stability by functioning as a crucial chain-terminating antioxidant [15]. Furthermore, in CD patients, the overrepresentation of module encoding for LPS synthesis may be linked to the overrepresentation of Enterobacteriaceae or other pathobionts [15]; indeed, Proteobacteria are considered the trigger of the inflammatory cascade [34]. In particular, normal intestinal mucosa contains resident commensal bacteria in the intact mucus layer, which also includes Proteobacteria. Epithelial cells, tight junctions, and the local immune system prevent the translocation of bacterial pathogens. However, potential changes in these protective systems, which can result from acute enteritis caused by pathogenic Proteobacteria, could be the primary trigger event in the pathogenesis of IBD. In this context, AIEC strains colonize the intestinal lumen, attach to the epithelial surface, and, subsequently, penetrate the lamina propria; they also proliferate within macrophage phagolysosomes and trigger the release of proinflammatory cytokines, which sustain the chronic inflammation typical of IBD patients [35]. The proinflammatory cytokines modify the luminal environment with resultant changes to the resident bacterial niche and the overgrowth of ‘inflammophilic’ pathogenic bacteria, which further aggravate the proinflammatory response [22]. AIEC strains exert their ‘inflammophilic’ effects through the activity of lipopolysaccharide (LPS), often referred to as endotoxin, a key structural component found in the external membrane of Gram-negative bacteria, such as *E. coli* [36]. LPS is known for its strong ability to stimulate the immune system and to trigger inflammation through several pathways: first, it binds to a receptor complex consisting of Toll-like receptor 4 (TLR4), CD14, Myeloid differentiation factor 2 (MD-2), and LPS-binding protein (LBP) found on innate immune cells, such as monocytes and macrophages [37]. This binding, activating the pathway of the nuclear factor NF-κB, leads to the release of proinflammatory cytokines, like IL-1β, IL-6, IL-12, and TNF-α [38]. Additionally, by binding to TLR4, LPS primes the NLRP3 inflammasome [39]. In humans, cytosolic LPS is directly detected by caspase-4 and caspase-5, which act as innate immune sensors, and their activation induces pyroptosis, a form of programmed cell death [40]. Moreover, LPS enhances the production of reactive oxygen species (ROS) in various cell types by activating NADPH oxidase [41]. Thus, through multiple pathways, LPS emerges as a pivotal mediator of inflammation in intestinal dysbiosis.

Interestingly, EEN treatment induces a marked reduction in the relative abundance of several bacterial species, including some that were represented in healthy controls: this finding is somewhat paradoxical, as it would be expected that EEN restores the “dysbiotic” microbiota toward a healthier composition while, on the contrary, EEN treatment appeared to drive the microbiota into an even more “dysbiotic” state [14,15]. Specifically, EEN induces a reduction in beneficial phyla (Firmicutes, in particular, *F. prausnitzii*) [42,43] as well as harmful phyla (Proteobacteria) [19,42]. The decline in *F. prausnitzii* is likely a consequence of reduced undigested food residues, which are essential for sustaining normal microbial populations in the distal gut [44]. This is not unexpected, given the composition of EEN formulas, which contain macronutrients, vitamins, minerals, trace elements, and readily digestible protein, with little or no fiber. However, the reduction in harmful Enterobacteriaceae mediated by EEN contributes to lowering the inflammatory trigger in CD [42,45]. Sustained remission in CD may be associated not only with decreased Proteobacteria but also with increased levels of *Akkermansia muciniphila* and Bacteroides [19].

In Table 1, the main mechanisms of EEN are summarized according to the targeted barrier.

### 1.2. Is It Possible to Enhance the Efficacy of EEN by Modulating Its Nutritional Composition?

The efficacy of EEN in inducing and maintaining remission has been demonstrated with different formulations (elemental, semi-elemental, and polymeric formulas) [11]. However, can the manipulation of the nutritional key ingredients enhance its clinical efficacy?

To answer this question, a narrative literature review to assess the effects of different enteral formulations in pediatric CD patients was performed. Although the primary focus of this review is on pediatric CD, relevant studies conducted in adult populations were also considered when they provided mechanistic insights or helped contextualize pediatric findings.

## 2. Materials and Methods

An extensive literature search was conducted across PubMed, Embase, and the Cochrane Library, covering all records published up to 27 July 2025. Grey literature sources, including conference proceedings and trial registries, were not considered. The search terms used included “enteral nutrition”, “Crohn’s disease”, “trial”, “pediatric”, “paediatric”, and “microbiota”. Additionally, the reference lists of all eligible articles were manually reviewed to identify further studies of interest. Studies were included if they met the following criteria: (a) clinical trials that used EEN for the induction of remission in CD; (b) the type of formula was clearly specified; and (c) complete remission data were available. The study selection process is illustrated in Figure 1. The nutritional values of the enteral formulations mentioned were obtained directly from the original texts and, when not available, from alternative sources, such as the official websites of the manufacturing companies. Studies on both pediatric and adult populations were considered for this literature review: the nutritional compositions of the enteral formulas has been analyzed in relation to remission rates to assess potential correlations.

## 3. Literature Review

Twelve studies involving patients with active CD were found. The studies varied in design, population, and nutritional formulas, but all of them assessed clinical remission and additional outcomes, such as inflammatory markers, nutritional status, endoscopy, histology, and, in two studies, gut microbiota.

Lochs et al. [46] conducted a randomized controlled trial (RCT) on 107 adults, comparing continuous EEN to a semi-elemental formula (Peptisorb^®^, Pfrimmer and Co., Erlangen, Germany) with steroids and sulfasalazine for 6 weeks. Remission was defined as a decrease in the initial Crohn’s Disease Activity Index (CDAI) by 40% or at least 100 points. After 6 weeks of therapy, there was a significant difference in remission rates between the two groups: among 55 patients treated with EEN, 29 people (52.7%) achieved clinical remission, and among the 52 patients treated with steroids, 41 participants (78.8%) reached remission. Both EEN and drugs led to a weight gain in responders: body weight increased from 55.6 ± 1.8 kg to 58.9 ± 1.6 kg in the diet group and from 53.5 ± 1.3 kg to 56.8 ± 1.2 kg in the drug group.

González-Huix et al. [47] studied 32 adult patients in an RCT, comparing 17 patients receiving steroids and 15 receiving continuous EEN based on a polymeric formula (Edanec HN^®^, UNIASA, Granada, Spain). Clinical remission was defined by clinical improvement and a Van Hees activity index (VHAI) of less than 120. At the end of the intervention, 88.2% (*n* = 15) of patients in the steroid group and 80% (*n* = 12) in the EN group achieved remission.

A non-randomized, open-label prospective study was conducted by Beattie et al. [48]. The study included 7 pediatric patients with active CD treated with a casein-based polymeric diet enriched with transforming growth factor β2 (TGFβ2 > 24 ppm; Nestlé-Clintec®, Vevey, Switzerland). Clinical improvement was observed in all patients, and histological remission was achieved in 2 of them (28.6%). A median weight gain of 5.2 kg was recorded, with weight increasing from 34 kg (range 28.3–45.2 kg) to 39.4 kg (range 29.8–49.1 kg).

Terrin et al. [49] designed an RCT that included 20 pediatric patients randomly assigned to receive either an 8-week course of steroids or an extensively hydrolyzed enteral formula (Pregomin^®^, Milupa). By week 8, 90% of patients in the EN group achieved clinical remission (PCDAI < 10), with a significant difference compared to 50% of the remission rate in the steroid group.

Sakurai et al. [50] randomized 36 patients assigned to receive either a low-fat elemental diet (Elental^®^, Ajinomoto Pharma, Tokyo, Japan) or a semi-elemental formula rich in MCTs (Twinline^®^, Otsuka Pharmaceutical, Tokyo, Japan) for 6 weeks. Short-term remission, defined as a ≥40% or ≥100-point reduction in CDAI, was achieved in 67% of the patients treated with the elemental diet and in 72% of the patients treated with the semi-elemental diet.

In a double-blind randomized controlled trial by Gassull et al. [51], 62 patients received, for a maximum of 4 weeks, a polymeric enteral formula high in oleate and low in linoleate and a polymeric formula with the opposite lipid profile, i.e., high in linoleate and low in oleate, with a remission rate of 20% and 52%, respectively. This finding underscores the possibility that the lipid profile may impact the therapeutic efficacy of EEN.

In a prospective, open-label randomized controlled trial, Borrelli et al. [52] compared the efficacy of a 10-week course of polymeric enteral nutrition (Modulen IBD^®^, Nestlé, Vevey, Switzerland) to oral steroids in children with active and treatment-naïve CD. By week 10, clinical remission (PCDAI ≤ 10) was achieved in 74% of patients in the EEN group and in 33% in the steroid group. Interestingly, only patients in the EEN group showed significant improvement in endoscopic and histologic scores, supporting the conclusion that polymeric enteral nutrition is more effective than steroids in inducing mucosal and histologic healing in newly diagnosed pediatric CD.

Rubio et al. [53] retrospectively studied 106 pediatric patients treated with Modulen IBD^®^ (Nestlé, Vevey, Switzerland) for an 8-week period. Patients were stratified into two groups according to the modality of administration: fractionated oral intake (oral group, *n* = 45) and continuous EEN via feeding tube (CEN group, *n* = 61). Clinical remission was defined as a PCDAI score < 10. At week 8, clinical remission was achieved in 75% of patients in the oral group and 85% in the CEN group. All patients demonstrated significant improvements in anthropometric measures, with differences in weight gain, which was greater in the CEN group. Follow-up endoscopy was performed in only 16 patients (15%), revealing mucosal healing in 75% (12/16), more frequently in the oral group (88%) than in the CEN group (63%). The authors concluded that no substantial differences exist between oral and continuous enteral administration of Modulen^®^ IBD, apart from greater weight gain observed with the enteral route.

Pigneur et al. [54] conducted a randomized controlled trial, comparing EEN for 8 weeks with Modulen IBD^®^ (Nestlé, Vevey, Switzerland) and steroids in 19 treatment-naive pediatric patients. Clinical remission was defined by the Harvey–Bradshaw Index (HBI), and mucosal healing was defined by the Crohn’s Disease Endoscopic Index of Severity (CDEIS). After 8 weeks, clinical remission was observed in 100% of patients in the EEN group and in 83% of the steroid group. Additionally, the study included an analysis of gut microbiota in a small subgroup at baseline and week 8. At baseline, no significant differences in gut microbiota composition were observed between the steroid-treated and EEN-treated groups, with Firmicutes (61%), Bacteroidetes (33%), Actinobacteria (3%), and Proteobacteria (1%) as the predominant phyla. After 8 weeks in the steroid group, there was an increase in *Ruminococcus* and butyrate-producing bacteria (e.g., *Roseburia intestinalis*, *Eubacterium*), alongside a significant reduction in *Blautia* and a trend toward increased *Bifidobacterium*, with only minimal changes in microbial diversity. In the EEN group, treatment was associated with increased microbial diversity (Shannon index 3.82→5.0), an expansion of *Clostridium cluster XIVa* and other species, such as *Ruminococcus torques* and *Clostridium hathewayi*, and a relative decrease in *Faecalibacterium* and *Roseburia* compared to the steroid group. Moreover, mucosal cytokine expression was analyzed in a subgroup of 11 patients (EN group: *n* = 8; steroid group: *n* = 3): no difference was found between the two groups.

Sigall Boneh et al. [55], with microbiota and metabolic analyses extended by Verburgt et al. [56], conducted a randomized controlled trial comparing EEN to the Crohn’s Disease Exclusion Diet (CDED) in 73 pediatric patients with active CD aged 4–18 years. Patients received either EEN (*n* = 34) or the CDED phase 1 diet plus PEN using Modulen IBD^®^ (*n* = 39) for 6 weeks. By the third week of dietary treatment, clinical response was achieved in 82% of patients receiving CDED and 85% of those treated with EEN. The microbiota analysis by Verburgt et al. [33] shows that successful dietary therapy shifted the gut microbiota of CD patients toward a healthier profile, with reduced Proteobacteria and increased Firmicutes. By week 12, most Proteobacteria genera were normalized, except *E. coli*. Although the SCFA synthesis pathways increased, this did not translate into higher SCFA levels: notably, in the EEN group, total and specific SCFAs, such as acetate, propionate, and butyrate, significantly declined during remission and then increased after dietary liberalization, while no significant SCFA changes were observed in the CDED group.

Dawson et al. [57] compared two cohorts of pediatric patients with active CD receiving EEN based on either a standard polymeric formula (Fortisip^®^, Nutricia, Danone, retrospective cohort) or a disease-specific polymeric formula (Modulen IBD^®^, Vevey, Switzerland, prospective cohort). A total of 171 patients were included (*n* = 106 Fortisip, *n* = 65 Modulen). The remission rate was similar in the two groups: 63% and 64%, respectively. Both groups demonstrated improvements in anthropometric parameters after EEN, although BMI z-scores at the end of therapy were slightly, but significantly, higher in the Modulen group (–0.1 vs. –0.5; *p* = 0.03).

The main results of each study are represented in Table 2. In Table 3, the detailed nutritional composition of the diet reported in each study is presented.

## 4. Discussion

The present review focused on the relationship between the nutritional composition of EEN diets and their clinical efficacy. Exclusive enteral nutrition is based on the exclusive use of elemental, semi-elemental, and polymeric diets [58]. Among these different types of EEN formulas, polymeric diets have become the favorite ones over time, probably due to their beneficial effects on clinical remission, higher palatability, and lower costs [59]. Indeed, the majority of the studies included in this review [47,48,51,52,53,54,55,57] utilized a polymeric formula.

Overall, five studies in this review employed a single EEN diet that is licensed and specifically marketed for CD, namely, a casein-based polymeric feed containing TGF-β, and more than fifty percent of the patients reached remission by the end of the treatment [52,53,54,55,57]. The favorable outcomes of polymeric diets containing milk proteins have been linked to the potential impact of TGF-β [26]. In fact, TGF-β is naturally present in milk and milk-based powders, predominantly as TGF-β2 [48,60,61]. This cytokine has been reported in experimental animals to be anti-inflammatory, specifically inhibiting IFN-γ expression on intestinal epithelial cells [48].

According to this narrative review, since semi-elemental and elemental diets without TGF-β showed similar remission rates (range 53–90%) [46,49,50], it can be hypothesized that the presence of TGF-β may enhance the anti-inflammatory properties of polymeric diets.

Specifically, TGF-β exists in five isoforms, sharing 60–80% homology and largely overlapping functions: among them, TGF-β1, -β2, and -β3 are expressed in mammals, including humans, and are produced by multiple cell types and organs, presenting primarily an immunosuppressive function [62]. In particular, the isoform TGF-β1 acts as an endogenous down-regulator of inflammation [26]. However, in CD mucosa, Smad7 is overexpressed and inhibits TGF-β1 signaling within cells [26]: indeed, Smad7 is detected in both lamina propria and epithelial cells of the terminal ileum in CD patients [63], thereby limiting the potential anti-inflammatory effects of TGF-β1. Consequently, the anti-inflammatory effects of casein-based EEN polymeric formulas may be enhanced by the presence of TGF-β, which may compensate for impaired TGF-β1 signaling in CD mucosa [48,60,61].

Another interesting feature in understanding the anti-inflammatory activity of EEN diets is their fat composition. Interestingly, in the polymeric EEN diet marketed for CD, fats provide 42% of the total calories, while carbohydrates provide 44% [52]. The fat-to-carbohydrate energy ratio is, therefore, quite different from that required for children aged 4 to 14 years [64]. Indeed, LARN recommendations for children aged 4–14 years indicate that carbohydrates and fats should provide 45–60% and 20–35% of the total energy intake, respectively [64]. The high fat-to-carbohydrate ratio may contribute to the clinical efficacy of EEN by modulating the balance between proinflammatory and anti-inflammatory effects, which depend more on the overall fat profile, rather than on a single fatty acid. So, the high amount of fat with a mixed composition, including saturated fatty acids (SFAs), monounsaturated fatty acids (MUFAs), polyunsaturated fatty acids (PUFAs) and medium-chain triglycerides (MCTs), has an anti-inflammatory, rather than nutritional, purpose [51]. However, the limited evidence from studies attempting to modify individual fat components should be interpreted cautiously. In this literature review, the enrichment with individual fat components does not appear to enhance the anti-inflammatory activity of EEN. Indeed, Sakurai et al. and Gassul et al. [50,51] tried to modify the ratio of omega-6 to omega-3 (ω6:ω3) to detect the efficacy of EEN diets in inducing remission and observed lower remission rates (Table 3). Likewise, the enrichment with MUFA failed to show higher anti-inflammatory activity compared to a mixed-fat composition [51]. Nevertheless, the small sample sizes of these studies and their methodological differences limit definitive conclusions about the optimal fat composition. Further research is needed to determine whether specific fatty acid profiles could enhance EEN efficacy, as the current evidence suggests that balanced mixed-fat compositions may be preferable to single-component enrichment strategies.

In addition, a relevant and surprising feature of EEN action in CD patients is represented by its ability to induce profound changes in the gut microbiota of CD patients, specifically in the microbiome diversity and in the *F. prausnitzii* levels, which had been depleted during EEN [14]. *F. prausnitzii*, belonging to the Firmicutes phylum, is a major representative of the *Clostridium leptum* group, which produces anti-inflammatory metabolites, including SCFAs, in particular, butyrate [14]. Butyrate is a known metabolic fuel for colonocytes, but it is also recognized for its anti-inflammatory activity, mainly through its capacity to alter the gene expression of MHC class II [6,30]. The available EEN formulas lack fibers [65]; yet, they exert good anti-inflammatory activity. While this suggests that the direct provision of SCFAs through dietary fiber may not be absolutely essential for the acute anti-inflammatory effects observed with EEN, it does not diminish the potential therapeutic value of SCFAs in IBD management. The clinical improvement, which was observed despite reduced *F. prausnitzii* andother Firmicutes strains, as well as decreased butyrate production [54,56], may indicate that the anti-inflammatory mechanisms of EEN operate through multiple pathways, some of which may be independent of SCFA-mediated effects. Therefore, it is possible that the fiber enrichment represents a strategy to enhance the anti-inflammatory activity of EEN diets by promoting the recolonization of eubiotic microbiota and the production of metabolites, such as SCFAs and butyrate [66,67,68,69].

Furthermore, in clinical practice, compliance with EEN is often limited by its taste and the high volumes required to reach individual caloric requirements. Therefore, modifies to EEN formulas, such as increasing caloric density to limit the required volume and improving palatability, could enhance adherence to this nutritional treatment [34].

Despite the relevance of these findings, some limitations of this review should be acknowledged. The main limitation of this review is the low number of studies included, which is a result of the chosen selection criteria. Indeed, we included only studies where patients were treated with EEN. This strategy was chosen to avoid confounding factors that could have prevented (a) an understanding of the role of EEN in treating the acute phases of CD and (b) a comparison of the different diets for detecting the potential role of the key ingredients they contain.

This review also has important strengths that should be underlined. First, it provides an in-depth examination of the pathophysiological mechanisms through which EEN exerts its therapeutic effects in pediatric CD, integrating microbiota-related changes. By examining the interplay between nutritional composition, intestinal inflammation, and gut microbial dynamics, the review provides a comprehensive understanding of how EEN influences disease activity. Second, it is the first work to relate the nutritional composition of different enteral formulas with reported clinical outcomes and remission rates, thereby providing novel insights to inform both clinical practice and future research.

In conclusion, EEN with polymeric diets satisfies the need of reverting the acute inflammation in most pediatric CD patients. Polymeric formulas have two advantages: (a) they contain TGF-β, which exerts anti-inflammatory effects on intestinal epithelial cells, and (b) they have a mixed-fat composition, including SFAs, MUFAs, PUFAs as well as MCTs, which may provide advantages compared to selective supplementation strategies, although clinical evidence remains limited and further research is needed to establish optimal fat profiles. However, pathophysiological evidence shows that EEN induces profound alterations in the gut microbiota, including reduced diversity and decreased *Faecalibacterium prausnitzii,* likely due to the absence of dietary fibers in EEN formulations. Although these changes may appear unfavorable, they occur in parallel with clinical remission, indicating a more complex relationship between microbial shifts and therapeutic benefits. So, a potential strategy to enhance the efficacy of polymeric diets could be fiber enrichment. Furthermore, in clinical practice, compliance with polymeric diets could be improved by increasing energy density and by improving palatability.

## Figures and Tables

**Figure 1 nutrients-17-03124-f001:**
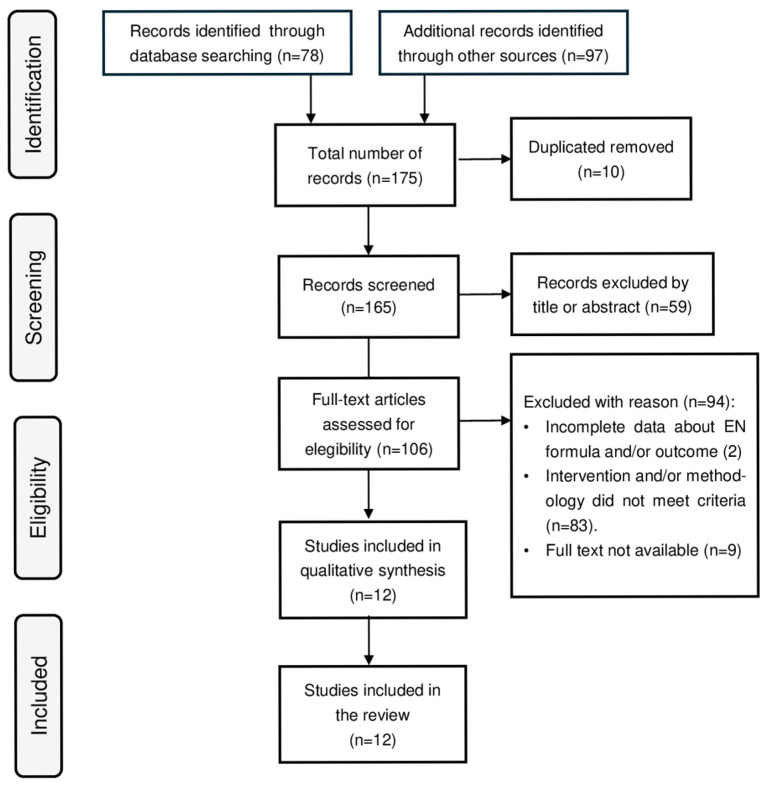
Flow diagram illustrating the study selection process.

**Table 1 nutrients-17-03124-t001:** Exclusive enteral nutrition (EEN) and Crohn’s disease (CD) pathophysiology. Comparison of key pathogenic mechanisms in Crohn’s disease (first column) with the corresponding therapeutic effects of exclusive enteral nutrition (second column).

	CD Pathophysiological Features	Specific EEN Actions
**Mucosal barrier** [5,9,10,11,12,13,14,15,16,17,18,19]	↓ Thickness of mucus layer	↑ Thickness of mucus layer by: Trophic effects on epithelial cells Modulation of dysbiosis (↑ *A. muciniphila*)
	↑ Epithelial layer permeability	↓ Epithelial layer permeability by: Inhibition of IL-1β, IL-6, IL-8, IFN-γ, and TNF-α Modulation of dysbiosis (↓ Proteobacteria) Trophic effects on epithelial cells
	↑ Tight junction permeability	↓ Tight junction permeability by: Inhibition of IL-1β, IL-6, IL-8, IFN-γ, and TNF-α Modulation of dysbiosis (↓ Proteobacteria) ↑ expression of tight junction proteins
**Immunological barrier** [5,9,20,21,22,23,24,25,26,27,28,29,30,31]	↑ Antigen presentation by dendritic cells, together with II MHC compatibility	↓ Capacity for antigen presentation via changes in gene expression of class II MHC
	↓ Treg	↑ Treg
	↑ Switch to Th1 response	↓ Switch to Th1 response by inhibition of IL-1β, IL-6, IL-8, IFN-γ, and TNF-α
	↓ Cell apoptosis	↑ Cells apoptosis by inhibition of IL-2, IL-15, IL-6, IL-12, and IL-18
**Microbiological barrier** [19,22,26,32,33,34,35,36,37,38,39,40,41,42,43,44,45]	↓ Firmicutes	↓ Firmicutes (*F. prausnitzii*) ↓ Proteobacteria (Enterobacteriaceae)
	↓ *Akkermansia muciniphila*	↑ *Akkermansia muciniphila*
	↑ AIEC strains	↓ AIEC strains

CD, Crohn’s disease. EEN, exclusive enteral nutrition. AIEC, adherent-invasive *Escherichia coli*. MHC, major histocompatibility complex. Treg, T-regulatory lymphocytes.

**Table 2 nutrients-17-03124-t002:** Demographic and laboratory data of enrolled patients after interventions. If *p*-values are not reported, they were not calculated/reported in the original articles.

References	Interventions	Patients	Age	Female (n, %)	CD Activity Score	Albumin	CRP and ESR
Lochs et al. [46]	EN (EN group) vs. corticosteroids and sulfasalazine (drug group)	Total: 107 patients EN group: 55 patients Drug group: 52 patients	Adult	EN group: 33, 60% Drug group: 37, 71%	CDAI < 150 within 6 weeks of interventions in: -43.6% (EN group) -67.3% (drug group)	Increased in both groups	N/A
González-Huix et al. [47]	EN (EN group) vs. corticosteroids (steroid group)	Total: 32 patients EN group:15 patients Steroid group: 17 patients	Adult	EN group: 8, 53.3% Steroid group: 7, 41.2%	Reduction in VHAI (%): -32.28% (EN group) -34.8% (steroid group)	Increased in: -26.7% (EN group) -29.4% (steroid group)	CRP decreased in: -20% (EN group) -23.5% (steroid group) ESR decreased in: -46.7% (EN group) -41.2% (steroid group)
Beattie et al. [48]	EN	Total: 7 patients	Pediatric	3, 42.9%	Significant improvement in LSI (*p* < 0.001)	Significantly increased (*p* < 0.001)	Both CRP and ESR decreased significantly (*p* < 0.001)
Terrin et al. [49]	EN (EN group) vs. corticosteroids (steroid group)	Total: 20 patients EN group: 10 patients Steroid group: 10 patients	Pediatric	N/A	Both treatments were effective in significantly reducing PCDAI, but only EN showed significantly lower post-intervention scores (*p* < 0.01 vs. *p* = NS)	Significantly increased only in the EN group (*p* < 0.01 vs. *p* = NS)	CRP decreased significantly in both groups, especially in the EN group (*p* < 0.01 vs. *p* < 0.05). ESR decreased significantly in both groups (*p* < 0.01 in both)
Sakurai et al. [50]	EN with elemental formula (ED group) vs. EN with semi-elemental formula (TL group)	Total: 36 patients ED group: 18 patients TL group: 18 patients	Young adults: ED group: 26.3 ± 8.0 years TL group: 25.3 ± 7.4 years	ED group: 4, 22.2% TL group: 2, 11.1%	Decreased in both groups. Over 6 weeks, CDAI decreased from 213 ± 8 to 102 in the ED group and from 195 ± 4.5 to 82 in the TL group	Increased in both groups, without significant differences	Both CRP and ESR decreased in both groups, without significant differences
Gassul et al. [51]	Polymeric enteral formula 1 (PEN1) vs. polymeric enteral formula 2 (PEN2) vs. corticosteroids (steroid group)	Total: 62 randomized patients (n = 44 PP) PEN1 group: 20 patients PEN2 group: 23 patients Steroid group: 19 patients	Adult	PEN1 group: 11, 55% PEN2 group: 13, 56.5% Steroid group: 10, 52.6%	Decreased in all groups, without significant differences between groups	Increased in all groups, without significant differences between groups	Both CRP and ESR decreased in all groups, without significant differences between groups
Borrelli et al. [52]	EN (EN group) vs. corticosteroids (steroid group)	Total: 37 randomized patients (n = 32 PP) EN group: 17 patients Steroid group: 15 patients	Pediatric	EN group: 12, 63.2% Steroid group: 10, 55.6%	PCDAI significantly decreased in each group (*p* < 0.001), with no differences between groups (*p* = NS)	Significantly increased in each group (*p* < 0.001), with no differences between groups (*p* = NS)	Both CRP and ESR decreased significantly in each group (*p* < 0.001), with no differences between groups (*p* = NS)
Rubio et al. [53]	Continuous EN (EN group) vs. oral fractionated EN (oral group)	Total = 106 patients EN group: 61 patients Oral group: 45 patients	Pediatric	EN group: 22, 36% Oral group: 14, 31%	PCDAI significantly decreased in both groups (*p* < 0.001), with no differences between groups (*p* = NS)	Increased in each group (*p* < 0.01), with no differences between groups (*p* = NS)	Both CRP and ESR decreased in each group (*p* < 0.01), with no differences between groups (*p* = NS)
Pigneur et al. [54]	EN (EN group) vs. corticosteroids (steroid group)	Total: 19 patients EN group: 13 patients Steroid group: 6 patients	Pediatric	4, 21%	HBI significantly improved, especially in the EN group (*p* < 0.05 compared to the steroid group)	Increased in each group, with no differences between groups (*p* = NS)	Both CRP and ESR decreased significantly in each group, with no differences between groups (*p* = NS)
Sigall Boneh et al. [55]	EN (EN group) vs. CDED (CDED group)	Total: 73 patients EN group: 34 patients CDED group: 39 patients	Pediatric	27, 37%	PCDAI significantly decreased in each group (*p* < 0.001)	N/A	CRP decreased significantly in each group (*p* < 0.001) ESR data N/A
Dawson et al. [57]	EN with two different polymeric formulas (Fortisip group vs. Modulen group)	Total: 171 patients Fortisip group: 106 patients Modulen group: 65 patients	Pediatric	70, 41%	N/A	N/A	No difference between the two groups in patients with normalization of CRP and ESR (*p* = NS) Fortisip patients had higher median CRP values compared to the Modulen group (*p* < 0.001)

CD, Crohn’s disease. CRP, C-reactive protein. ESR, erythrocyte sedimentation rate. EN, enteral nutrition. VS, versus. CDAI, Crohn’s Disease Activity Index. VHAI, Van Hees activity index. LSI, Lloyd Still Index. PCDAI, Pediatric Crohn’s Disease Activity Index. NS, not significant. HBI, Harvey–Bradshaw Index. CDED, Crohn’s Disease Exclusion Diet. N/A, not available.

**Table 3 nutrients-17-03124-t003:** Nutritional composition of exclusive enteral nutrition (EEN) diets according to the remission rate.

Reference	Rem. Rate (%)	P.	S.	E.	Protein	Type of Protein	Fat	SFAs	MCTs	MUFAs	ω6	ω3	ω6:ω3	CHO	Glucose Polymers
Terrin et al. [49]	90		X		1.8	Whey	3.4	2.1	0	0.7	0.44	0.054	8.1:1	7.2	6.1
Rubio et al. [53]	85	X			3.5	Casein	4.6	2.6	1.2	0.78	0.42	0.004	10:1	10.8	4.6
Sigal Bonneh et al. [55]	85	X			3.5	Casein	4.6	2.6	1.2	0.78	0.42	0.004	10:1	10.8	4.6
González-Huix et al. [47]	80	X			5.5	NR	3.6	1	0.5	1.47	NR	NR	NR	11.4	/
Borrelli et al. [52]	74	X			3.5	Casein	4.6	2.6	1.2	0.78	0.42	0.004	10:1	10.8	4.6
Sakurai et al. [50]	72			X	3.38	/	0.3	/	/	/	0.09	0.015	6:1	163	NR
Sakurai et al. [50]	67		X		3.2	NR	5	/	2	/	1.3	/	/	11.8	/
Dawson et al. [57]	64	X			3.5	Casein	4.6	2.6	1.2	0.78	0.42	0.004	10:1	10.8	4.6
Gassul et al. [51]	63	X			5.4	Casein	3.3	0.34	0.19	2.6	0.21	0.05	4.2:1	11.6	11.6
Dawson et al. [57]	63	X			5.9	Casein	5.8	0.6	/	3.5	0.7	0.14	5:1	18.4	11.7
Pigneur et al. [54]	62	X			3.5	Casein	4.6	2.6	1.2	0.78	0.42	0.004	10:1	10.8	4.6
Lochs et al. [46]	53		X		2.8	Whey	3.9	2.2	1.8	0.5	1.02	0.093	10.9:1	13.7	12
Beattie et al. [48]	29	X			2.8	Casein	6.25	3.8	NR	NR	0.7	NR	NR	10.8	10.8
Gassul et al. [51]	27	X			5.4	Casein	3.3	0.55	0.26	0.93	1.5	0.05	30:1	11.6	11.6

Nutritional composition is reported as g/100 mL. P.: polymeric formula; S.: semi-elemental formula; E: elemental formula; ω6: omega-6 fatty acids; ω3: omega-3 fatty acids; MUFAs, monounsaturated fatty acids; MCTs, medium-chain triglycerides; CHO: carbohydrates; NR: not reported; SFAs: saturated fatty acids.

## Data Availability

No new data were created or analyzed in this study. Data sharing is not applicable to this article.

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
