# Peer review of "Enteral Nutrition in Pediatric Crohn’s Disease: New Perspectives"

_nutrients, 2025, doi:10.3390/nu17193124_

Round 1

Reviewer 1 Report

Comments and Suggestions for Authors

General comments:

The entire manuscript is a compilation of random phrases in which each statement is unrelated to the next, and there is no reasonable thread running through the narrative. It is impossible to read a paragraph and continue to understand what the reader is reading. It could easily be a text produced by AI. The tables are completely out of format, and the review does not really add anything new.

Specific comments:

Abstract: There are a lot of abbreviations that are only cited once. It does not make sense to abbreviate if it is not cited subsequently, such as “SFA”, “MCT”, etc.

Line 18: “we planned” if previously it was redacted in passive voice, please maintain the same redaction style.

“LPS” was first cited in line 163 and defined in line 177.

Tables are out of required format

References are out of required format

All the main text are a compilation of subsequent phrased without a main theme of the text

Comments on the Quality of English Language

All the main text are a compilation of subsequent phrased without a main theme of the text. It seems be redacted by an IA.

Author Response

Comments 1: General comments: The entire manuscript is a compilation of random phrases in which each statement is unrelated to the next, and there is no reasonable thread running through the narrative. It is impossible to read a paragraph and continue to understand what the reader is reading. It could easily be a text produced by AI. The tables are completely out of format, and the review does not really add anything new.

Response 1: We thank the Reviewer for this critical and constructive assessment. The manuscript has undergone a thorough, line-by-line edit to improve coherence, paragraph structure, and argument flow. Specifically, we (i) merged fragmented statements into logically connected paragraphs throughout the manuscript (Introduction: page 1, lines 37-40; page 2, lines 55, 58, 72-75; page 4 lines 164, 171, 172; page 5, line 199; Discussion: page 16, lines 380-384,390, 397, 402, 419, 424, 431, 445, 462-465); (ii) tables have been fully reformatted to conform to Nutrients style (the text size has been reduced to 8, as recommended by the Nutrients Instructions for Authors (https://www.mdpi.com/journal/nutrients/instructions), and we have adjusted the table dimensions to fit the page as much as possible, in particular tables 2 and 3 have been formatted in horizontal layout to improve clarity and readability, given the large number of columns (Tables 1-3, pages 6, 11-15); (iii) for greater clarity, the methods have been further specified by adding the following sentence at the end of the Introduction (page 6, lines 230-234): “although the primary focus of this review is on pediatric CD, relevant studies conducted in adult populations were also considered when they provided mechanistic insights or helped contextualize pediatric findings”. Consistently, the abstract now includes brief details about the experimental design and data analysis in the Methods section: “An extensive literature search was performed across PubMed, Embase, and the Cochrane Library, covering all records published up to July 27, 2025. Both pediatric and adult studies were considered, and nutritional composition was correlated with remission rates.” (page 1, lines 19-22). The novel contributions and strengths of this review have already been clarified in the final part of the Discussion, prior to the Conclusion, where we stated that: “First, it provides an in-depth examination of the pathophysiological mechanisms through which EEN exerts its therapeutic effects in pediatric CD, integrating microbiota-related changes. By examining the interplay between nutritional composition, intestinal inflammation, and gut microbial dynamics, the review provides a comprehensive understanding of how EEN influences disease activity. Second, it is the first work to relate the nutritional composition of different enteral formulas with reported clinical outcomes and remission rates, thereby providing novel insights to inform both clinical practice and future research” (page 17, line 462-465). To enhance precision and clarity, the results were further specified both in the Abstract and in the main text. In the Abstract, the following sentence was added: “Most studies were conducted with polymeric diets (n=8), which achieved a high remission rate (up to 85%), thus confirming their advantage over other EEN diets” (page 1, lines 23-25). In the Discussion, we included the sentence: “Indeed, the majority of the studies included in this review [47,48, 51–55, 57] utilized a polymeric formula” (page 16, lines 384-385). Finally, we declare that we have not used AI platforms for writing the manuscript.

Comments 2: Abstract: There are a lot of abbreviations that are only cited once. It does not make sense to abbreviate if it is not cited subsequently, such as “SFA”, “MCT”, etc.

Response 2: We respectfully note that all abbreviations used in the Abstract were already defined upon their first appearance. For example, on page 1, lines 15–16, the terms “Exclusive Enteral Nutrition (EEN) and “Crohn’s Disease (CD)” are explicitly introduced before their abbreviations. Similarly, at lines 28–31, “transforming growth factor-β (TGF-β)”, “saturated fatty acids (SFA)”, “monounsaturated fatty acids (MUFA)”, “medium-chain triglycerides (MCT)”, and “polyunsaturated fatty acids (PUFA)” have been spelled out prior to the abbreviation. No abbreviation was used without an accompanying explanation. We have nevertheless carefully re-checked the Abstract to ensure clarity and consistency.

Comments 3: Line 18: “we planned” if previously it was redacted in passive voice, please maintain the same redaction style.

Response 3: Thank you. We have adopted an impersonal, passive construction throughout. In the Methods section of the Abstract, “we planned” has been changed to “a narrative review was conducted” ensuring stylistic consistency. (page 1, Abstract, line 17). Moreover, we have revised the manuscript to minimize first-person pronouns throughout (Introduction, page 2, lines 72-74; page 5, line 200; page 6, lines 225, 228, 230-231; Literature Review, page 10, line 356; Discussion, page 17, line 455).

Comments 4: “LPS” was first cited in line 163 and defined in line 177.

Response 4: We appreciate this observation. “Lipopolysaccharide (LPS)” is now defined at first mention (page 5, line 185), and subsequent occurrences use the abbreviation only. We also checked all sections to ensure that each abbreviation is expanded upon first appearance within the main text and tables.

Comments 5: Tables are out of required format

Response 5: We appreciate the Reviewer’s remark. All tables have now been revised and reformatted strictly according to the Nutrients Instructions for Authors (https://www.mdpi.com/journal/nutrients/instructions). In particular, each table was created using the “Table” function in Microsoft Word, with clear explanatory headings for every column, a short descriptive title, and a complete caption including explanations of symbols and abbreviations. Table order has been checked against their first appearance in the text, and references are now consistently presented in square brackets. Moreover, the text size in tables 1-3 has been reduced to 8, as recommended by the Nutrients Instructions for Authors, and we have adjusted the table dimensions to fit the page as much as possible, in particular tables 2 and 3 have been formatted in horizontal layout to improve clarity and readability, given the large number of columns (Tables 1-3, pages 6, 11-15). These adjustments ensure that the tables fully comply with the editorial guidelines on structure, clarity, and formatting.

Comments 6: References are out of required format

Response 6: We thank the Reviewer for raising this point. In accordance with the journal’s “Free Format Submission” policy (Instructions for Authors available at https://www.mdpi.com/journal/nutrients/instructions), references were initially provided in a consistent free format. For this revision, the entire reference list has been harmonized using the AMA (American Medical Association) style, applied consistentl. For example, the first reference is written as “Gordon H, Minozzi S, Kopylov U, et al. ECCO Guidelines on Therapeutics in Crohn’s Disease: Medical Treatment. J Crohns Colitis. 2024;18(10):1531-1555. doi:10.1093/ecco-jcc/jjae091.”

Comments 7: All the main text are a compilation of subsequent phrased without a main theme of the text

Response 7: We respectfully acknowledge the Reviewer’s concern. We would like to underline that the manuscript already followed a logical structure: it begins with the Introduction, proceeds with the pathophysiological mechanisms of EEN (organized into mucosal, immunological, and microbiological barriers), continues with a review of clinical trials including nutritional composition of formulas, and concludes with the Discussion. A coherent thematic thread was therefore present. Nevertheless, we recognized that in several sections the writing style relied on short, segmented statements, which might have reduced the perceived fluidity. To address this, we have restructured the narrative to enhance readability (Introduction: page 1, lines 37-40; page 2, lines 55, 58, 72-75; page 4 lines 164, 171, 172; page 5, line 199; Discussion: page 16, lines 380-384,390, 397, 402, 419, 424, 431, 445, 462-465).

Comments 8: Comments on the Quality of English Language: All the main text are a compilation of subsequent phrased without a main theme of the text. It seems be redacted by an IA.

Response 8: We acknowledge the Reviewer’s observation regarding the quality of the English language. We would like to emphasize that the manuscript was originally drafted in an academic style, with technical terminology and adherence to scientific conventions. However, we recognize that certain sections relied on short, segmented sentences, which may have given an impression of reduced fluency. To further improve readability, the text has now been carefully refined: overly long sentences were simplified, fragmented statements were merged, and transitional phrases were added to enhance narrative flow (Introduction: page 1, lines 37-40; page 2, lines 55, 58, 72-75; page 4 lines 164, 171, 172; page 5, line 199; Discussion: page 16, lines 380-384,390, 397, 402, 419, 424, 431, 445, 462-465). In addition, the manuscript has undergone professional English language editing to ensure clarity, conciseness, and consistency with academic standards. Finally we declare that we have not used AI platforms for writing the manuscript.

Reviewer 2 Report

Comments and Suggestions for Authors

The manuscript titled "Enteral Nutrition in Pediatric Crohn’s Disease: New Perspectives" provides a valuable and comprehensive review of the use of Exclusive Enteral Nutrition in managing acute phases of Crohn’s disease in pediatric patients. One of its key strengths is the detailed exploration of the pathophysiological mechanisms behind EEN’s therapeutic effects, particularly the role of gut microbiota changes. It is the first review to correlate the nutritional composition of different enteral formulas with clinical outcomes and remission rates, offering novel insights for both clinical practice and future research. Although the limited number of included studies is a clear limitation, as acknowledged by the authors, the strict inclusion criteria help maintain focus and reduce confounding factors. The structure of the manuscript is logical, and the flow of information is easy to follow. The language is clear and concise, which effectively communicates complex scientific concepts.

However, I have a few comments:

  1. the graphic abstract is missing
  2. rebuild firts sentence of the introduction (or divide it into 2)
  3. in table 1, write the reference to the literature in square brackets
  4. Figure 1 - place the description of the figure below the figure
  5. Table 3- remove expression: Table 3 Legend
  6. Discussion- rebuild 1 paragraph, divide into smaller sentences for better understanding

As already mentioned, the biggest limitation of the article is the number  of articles (12) on which the main theses of this review are based. 

Author Response

The manuscript titled "Enteral Nutrition in Pediatric Crohn’s Disease: New Perspectives" provides a valuable and comprehensive review of the use of Exclusive Enteral Nutrition in managing acute phases of Crohn’s disease in pediatric patients. One of its key strengths is the detailed exploration of the pathophysiological mechanisms behind EEN’s therapeutic effects, particularly the role of gut microbiota changes. It is the first review to correlate the nutritional composition of different enteral formulas with clinical outcomes and remission rates, offering novel insights for both clinical practice and future research. Although the limited number of included studies is a clear limitation, as acknowledged by the authors, the strict inclusion criteria help maintain focus and reduce confounding factors. The structure of the manuscript is logical, and the flow of information is easy to follow. The language is clear and concise, which effectively communicates complex scientific concepts.

However, I have a few comments:

Comment 1: the graphic abstract is missing

Response 1: We thank the Reviewer for this observation. A graphic abstract has now been prepared and included in the revised submission, in order to provide a visual overview of the review content.

Comment 2: rebuild firts sentence of the introduction (or divide it into 2)

Response 2: Thank you for this suggestion. The first sentence of the Introduction has been divided into two shorter statements for clarity and readability. The revised version is: “Exclusive enteral nutrition (EEN) consists of a complete liquid diet that meets all nutritional requirements while excluding all regular solid foods for a duration of 6 to 8 weeks [1]. This type of nutritional intervention is recommended as induction therapy to achieve remission in Crohn’s disease (CD) [1].” (page 1, Introduction, lines 37-40).

Comment 3:  in table 1, write the reference to the literature in square brackets

Response 3: Thank you for noticing it, this has been corrected (page 6, Table 1).

Comment 4: Figure 1 - place the description of the figure below the figure

Response 4: Thank you for this suggestion, the description of the Figure 1 has been placed below the figure (page 7, Figure 1).

Comment 5: Table 3- remove expression: Table 3 Legend

Response 5: Thank you, “Table 3 Legend” has been removed from the footnotes of Table 3 (page 15, Table 3).

Comment 6: Discussion- rebuild 1 paragraph, divide into smaller sentences for better understanding

Response 6: We thank the Reviewer for this suggestion. The first paragraph of the Discussion has been revised as follows: “The present review focused on the relationship between the nutritional composition of EEN diets and their clinical efficacy. EEN is based on exclusive use of elemental, semi-elemental, and polymeric diets [58]. Among these different type of EEN formula, polymeric diets have become over time the favorite ones, probably due to their beneficial effects on clinical remission, higher palatability and lower costs [59]. Indeed, the majority of the studies included in this review [47,48, 51-55, 57] utilized a polymeric formula.” (Discussion, page 16, lines 380-385).

Comment 7: As already mentioned, the biggest limitation of the article is the number  of articles (12) on which the main theses of this review are based.

Response 7: We thank the Reviewer for this critical observation. We are fully aware that the limited number of studies (n=12) included in this review represents a major limitation. However, this was the direct result of the strict inclusion criteria applied in the methodology. In particular, we deliberately excluded studies in which exclusive enteral nutrition was administered concomitantly with other therapies (e.g., immunosuppressive agents, even at low doses), in order to evaluate solely the independent effect of enteral nutrition diets and their composition. This choice, while reducing the number of eligible articles, was intended to minimize confounding factors and strengthen the focus of the review. We have already highlighted this limitation in the Discussion (Discussion, page 17, lines 453-459).

Reviewer 3 Report

Comments and Suggestions for Authors

Enteral Nutrition in Pediatric Crohn’s Disease: New Perspectives

Abstract and everywhere else: minimize the use of 'WE' and 'OUR'. Provide brief details about experimental design and data analysis. 

Line 41: replace "they"

L41-43: revise, not clear. 

L63-68: add proper references

L95 and everywhere else: Don't start the sentence with an abbreviation.

L211: revise

L244 and everywhere else: I suggest removing of P value, it's enough to mention that there was a significant difference, or highly significant, etc.

Discussion and conclusion: no comments

Author Response

Enteral Nutrition in Pediatric Crohn’s Disease: New Perspectives

Comment 1: Abstract and everywhere else: minimize the use of 'WE' and 'OUR'. Provide brief details about experimental design and data analysis.

Response 1: We have revised the manuscript to minimize first-person pronouns throughout (Abstract, page 1, lines 17-18; Introduction, page 2, lines 72-74; page 5, line 200; page 6, lines 225, 228, 230-231; Literature Review, page 10, line 356; Discussion, page 17, line 455).

The abstract now includes brief details about the experimental design and data analysis in the Methods section: “An extensive literature search was performed across PubMed, Embase, and the Cochrane Library, covering all records published up to July 27, 2025. Both pediatric and adult studies were considered, and nutritional composition was correlated with remission rates.” (Abstract, page 1, lines 19-22; Material and Methods, page 7, lines 248-250).

Comment 2: Line 41: replace "they"

Response 2: Thank you, this has been corrected: “In 1973, Voitk and coworkers [2] reported the first successful experience with EEN, administered as an exclusive elemental diet for an average of 22 days, in 12 of the 13 treated patients. The diet was well tolerated and led to weight gain and reduced inflammation; moreover, 2 of the 9 patients awaiting surgery no longer required it after the nutritional intervention” (Introduction, pages 1-2, lines 40-44).

Comment 3:  L41-43: revise, not clear. 

Response 3: We revised the sentences as follows: “In 1973, Voitk and coworkers [2] reported the first successful experience with EEN, administered as an exclusive elemental diet for an average of 22 days, in 12 of the 13 treated patients. The diet was well tolerated and led to weight gain and reduced inflammation; moreover, 2 of the 9 patients awaiting surgery no longer required it after the nutritional intervention” (Introduction, pages 1-2, lines 40-44).

Comment 4: L63-68: add proper references

Response 4: Thank you for noting this, which has been corrected. “Since the 1980, it has been known that EEN has a direct anti-inflammatory effect on the intestine in CD, mediated by improved intestinal permeability. Sanderson et al. [8] studied intestinal permeability to sugar to assess the efficacy of an elemental diet as an exclusive treatment of the small bowel CD. Fourteen children aged 11-17 years with active CD received an elemental diet for six weeks: tests with iso-osmolar oral test solutions before and after this treatment demonstrated abnormally elevated lactulose/rhamnose permeability ratios in all 14 children, which fell significantly after EEN beginning [8]. This change overlapped with marked clinical improvement, as assessed by a disease activity index score [8]” (Introduction, page 2, lines 63-71).

Comment 5: L95 and everywhere else: Don't start the sentence with an abbreviation.

Response 5: Thank you for signaling this, every sentence with an abbreviation at its beginning has been corrected (Introduction: page 3, lines 99-100, 102, 104; page 4, lines 146, 149-150, 164, page 5, line 199).

Comment 6: L211: revise

Response 6: The sentences in lines 209–212 of the Introduction have been revised to improve fluency and accuracy, as follows: “However, the reduction of harmful Enterobacteriaceae mediated by EEN contributes to lowering the inflammatory trigger in CD [42,45]. Sustained remission in CD may be associated not only with decreased Proteobacteria but also with increased levels of Akkermansia muciniphila and Bacteroides [19].”

Comment 7: L244 and everywhere else: I suggest removing of P value, it's enough to mention that there was a significant difference, or highly significant, etc.

Response 7: Thank you for this suggestion, p values have been removed throughout the main text, whereas statistical significance was expressed (page 8, lines 263-264, 284; page 9, lines 310-312; page 10, line 354).

Comments 8: Discussion and conclusion: no comments

Response 8: We thank the Reviewer for this positive feedback and are pleased that the Discussion and Conclusion were found to be appropriate.

Reviewer 4 Report

Comments and Suggestions for Authors

Comments to the Authors of manuscript number nutrients-3866060 entitled “Enteral Nutrition in Pediatric Crohn’s Disease: New Perspectives” Here is a detailed assessment of the problematic sections of the manuscript, including line breaks and explanations of why they are illogical, incorrect, or lacking in substance:

  1. Commercial polymer diets used in pediatric Crohn’s disease do not contain pharmacologically significant amounts of TGF-β. This claim is incorrect and may mislead the reader.
  2. “They have a mixed fat composition... which produces better results than EEN supplemented with single fat components.” This is an overly generalized statement, lacking solid clinical evidence.
  3. “Dysbiosis significantly worsens after initiating EEN due to a lack of fiber...” Not entirely correct. EEN causes profound changes in the microbiota (decreased diversity, decreased Faecalibacterium), but linking this unequivocally to “worsening dysbiosis” is simplistic. Moreover, clinically, the effect of EEN is positive.
  4. Table: Contradictory effects cannot be stated in the same table. The authors should indicate that the study results are inconsistent.
  5. "The polymer diet does not contain fiber, but it has anti-inflammatory effects, suggesting that butyrate is not essential." This is a simplification and a non-substantive conclusion. The lack of fiber does not mean that SCFAs are unimportant.
  6. Lines 424–427: Repeating the error from the previous fragment (p1). The comparisons regarding fats are too categorical.

Author Response

Comments to the Authors of manuscript number nutrients-3866060 entitled “Enteral Nutrition in Pediatric Crohn’s Disease: New Perspectives” Here is a detailed assessment of the problematic sections of the manuscript, including line breaks and explanations of why they are illogical, incorrect, or lacking in substance:

Comment 1: Commercial polymer diets used in pediatric Crohn’s disease do not contain pharmacologically significant amounts of TGF-β. This claim is incorrect and may mislead the reader.

Response 1: We thank the Reviewer for this important clarification. We have carefully revised the text. Specifically, in the Abstract the expression “they are rich in transforming growth factor-β (TGF-β)” has been changed to “they contain” (page 1, line 27). On page 16, line 387, “a casein-based polymeric feed rich in TGF-β” has been changed to “a casein-based polymeric feed containing TGF-β”. Similarly, on page 17, lines 469–470, “they are rich in TGF-β that exerts anti-inflammatory effects on intestinal epithelial cells” has been revised to “they contain TGF-β”. In addition, where available, we had already reported the exact amounts of TGF-β present in specific enteral formulas, so as to provide accurate, evidence-based information without overinterpretation.

Comment 2: “They have a mixed fat composition... which produces better results than EEN supplemented with single fat components.” This is an overly generalized statement, lacking solid clinical evidence.

Response 2: We acknowledge this overgeneralization. The text has been revised: “they have a mixed fat composition including SFA, MUFA, MCT, and PUFA, which may provide advantages compared with selective supplementation strategies, although clinical evidence remains limited and further research is needed to establish optimal fat profiles”. (page 18, lines 471-474)

Comment 3: “Dysbiosis significantly worsens after initiating EEN due to a lack of fiber...” Not entirely correct. EEN causes profound changes in the microbiota (decreased diversity, decreased Faecalibacterium), but linking this unequivocally to “worsening dysbiosis” is simplistic. Moreover, clinically, the effect of EEN is positive.

Response 3: We thank the Reviewer for this important observation which is crucial to avoid misunderstanding. The statement has been revised and now specifies that “EEN induces profound alterations in gut microbiota, including reduced diversity and decreased Faecalibacterium prausnitzii, likely due to the absence of dietary fibers in EEN formulations. Although these changes may appear unfavorable, they occur in parallel with clinical remission, indicating a more complex relationship between microbial shifts and therapeutic benefit” (page 18, lines 474-478). Moreover we revised the other parts of the main text regarding EEN and dysbiosis. At page 17, in the Discussion, another statement has been revised (“A relevant and surprising feature of EEN action in CD patients is represented by its attitude to worsen the dysbiosis in CD patients and specifically the microbiome diversity and the F. prausnitzii levels, which had been depleted during EEN [14]” and changed to “In addition, a relevant and surprising feature of EEN action in CD patients is represented by its attitude to induce profound changes in the gut microbiota of CD patients, specifically in the microbiome diversity and in the F. prausnitzii levels, which had been depleted during EEN [14]” (page 17, lines 431-434). Consistently, the statement “however, pathophysiological evidence shows that dysbiosis markedly worsens after EEN beginning, due to the lack of fibers in these diets.” of the Abstract, page 1, lines 32-34, has been changed to “however, pathophysiological evidence shows gut microbiota alterations after EEN beginning, despite clinical improvement. So, a potential strategy to enhance the efficacy of polymeric diets may be the fiber enrichment”. This point has been also addressed and further explored in the Introduction (page 5, lines 199-212): “Interestingly, EEN treatment induces a marked reduction in the relative abundance of several bacterial species, including some that were represented in healthy controls: this finding results somewhat paradoxical, as it would be expected that EEN restore the “dysbiotic” microbiota toward a healthier composition while, on the contrary, EEN treatment appeared to drive the microbiota into an even more “dysbiotic” state [14, 15]. In detail, EEN induces a reduction in beneficial phyla (Firmicutes, in particular F. prausnitzii) [42, 43] but also harmful phyla (Proteobacteria) [19, 42]. The decline in F. prausnitzii is likely a consequence of reduced undigested food residues, which are essential for sustaining normal microbial populations in the distal gut [44]. This is not unexpected, given the composition of EEN formulas, which contain macronutrients, vitamins, minerals, trace elements, and readily digestible protein, with little or no fibers. However, the reduction of harmful Enterobacteriaceae mediated by EEN contributes to lowering the inflammatory trigger in CD [42,45]. Sustained remission in CD may be associated not only with decreased Proteobacteria but also with increased levels of Akkermansia muciniphila and Bacteroides [19].”

Comment 4: Table: Contradictory effects cannot be stated in the same table. The authors should indicate that the study results are inconsistent.

Response 4: -- We thank the Reviewer for this comment and the opportunity to clarify the structure of Table 1. The table does not present contradictory findings from the same studies. Rather, its purpose is to provide a comparative overview: in the first column, we summarize the key pathophysiological mechanisms that drive Crohn’s disease, while in the second column we illustrate the corresponding therapeutic mechanisms through which exclusive enteral nutrition exerts its beneficial effects. These are therefore two distinct levels of evaluation placed side by side to highlight how EEN addresses the underlying pathogenic features of the disease. To prevent possible misunderstanding, we have revised the table caption and added a clarifying note: “Comparison of key pathogenic mechanisms in Crohn’s disease (first column) with the corresponding therapeutic effects of exclusive enteral nutrition (second column).” (page 6, Table 1 caption, lines 218-220).

Comment 5: "The polymer diet does not contain fiber, but it has anti-inflammatory effects, suggesting that butyrate is not essential." This is a simplification and a non-substantive conclusion. The lack of fiber does not mean that SCFAs are unimportant.

Response 5: We acknowledge the reviewer's valid concern about our oversimplified statement. The comment correctly points out that the absence of fiber in polymeric diets does not diminish the potential importance of SCFAs in inflammatory bowel disease management. We propose the following revision (page 17, lines 438-448):

"The available polymeric diet lacks fibers, yet it exerts good anti-inflammatory activity. While this suggests that the direct provision of SCFAs through dietary fiber may not be absolutely essential for the acute anti-inflammatory effects observed with EEN, it does not diminish the potential therapeutic value of SCFAs in IBD management. The clinical improvement, which was observed despite reduced F. prausnitzii and butyrate production, may indicate that the anti-inflammatory mechanisms of EEN operate through multiple pathways, some of which may be independent of SCFA-mediated effects. Therefore, it is not possible to exclude that the fiber enrichment could represent a strategy to enhance the anti-inflammatory activity of EEN diets by promoting the recolonization with eubiotic microbiota and the production of metabolites such as SCFAs and butyrate [66-69]. "

Comment 6: Lines 424–427: Repeating the error from the previous fragment (p1). The comparisons regarding fats are too categorical.

Response 6: We agree with the reviewer's assessment that our comparisons regarding fats were overly categorical and lacked nuance. We propose a revised version in the Discussion around fat composition (pages 16-17, lines 419-430) to be more measured:

"However, the limited evidence from studies attempting to modify individual fat components should be interpreted cautiously. In this literature review the enrichment with individual fat components does not appear to enhance the anti-inflammatory activity of EEN. Indeed, Sakurai et al. and Gassul et al. [50, 51] tried to modify the omega-6 to omega-3 (ω6:ω3) ratio to detect the efficacy of EEN diets in inducing remission, finding lower remission rates (Table 3). Likewise, the enrichment with MUFA failed to show higher anti-inflammatory activity if compared to a mixed fat composition [51]. Nevertheless, the small sample sizes of these studies and their methodological differences limit definitive conclusions about optimal fat composition. Further research is needed to determine whether specific fatty acid profiles could enhance EEN efficacy, as the current evidence suggests that balanced, mixed fat compositions may be preferable to single-component enrichment strategies”.

Round 2

Reviewer 1 Report

Comments and Suggestions for Authors

All my previous comments were satisfactorely answered and the manuscript was strongly improved with respect to the original version of the manuscript. Now, it is aceptable for its publication.